# Citrate-modified bacterial cellulose as a potential scaffolding material for bone tissue regeneration

**Rabiu Salihu** [1,2,3]*, **Saiful Izwan Abd Razak** [1,4], **Mohd Helmi Sani** [2], **Mohammed Ahmad Wsoo** [5], **Nurliyana Ahmad Zawawi** [2], **Shafinaz Shahir** [2]

**1** Centre for Advanced Composite Materials, Universiti Teknologi Malaysia, Skudai, Johor, Malaysia, **2** Faculty of Science, Department of Biosciences, Universiti Teknologi Malaysia, Skudai, Johor, Malaysia, **3** Department of Microbiology and Biotechnology, Federal University Dutse, Dutse, Jigawa, Nigeria, **4** Faculty of Engineering, Bioinspired Device and Tissue Engineering Research Group, School of Biomedical Engineering and Health Sciences, Universiti Teknologi Malaysia, Skudai, Johor, Malaysia, **5** Department of Chemistry, College of Science, University of Raparin, Ranya, Kurdistan Region, Iraq

* salihu.r@fud.edu.ng

## Abstract

Bacterial cellulose (BC) is a novel biocompatible polymeric biomaterial with a wide range of biomedical uses, like tissue engineering (TE) scaffolds, wound dressings, and drug delivery. Although BC lacks good cell adhesion due to limited functionality, its tunable surface chemistry still holds promise. Here, hydroxyapatite (HA) was incorporated into a citrate-modified BC (MBC) using the biomimetic synthesis in simulated body fluid (SBF). Fourier transform infrared (FTIR) spectroscopy, X-ray diffraction (XRD), field emission scanning electron microscopy (FE-SEM), thermal gravimetric analysis (TGA), and compressive modulus were used to characterize the biomineralized MBC (BMBC) samples. Using 3-(4,5 dimethylthiazol-2-yl)-5-(3-carboxymethoxyphenyl)-2-(4-sulfophenyl) -2H-tetrazolium (MTS), trypan blue dye exclusion (TBDE), and cell attachment assays on osteoblast cells, the developed BMBC have shown good cell viability, proliferation, and attachment after 3, 5, and 7 days of culture and therefore suggested as potential bone tissue regeneration scaffolding material.

## Introduction

Diseases, injuries, and trauma were the significant causes of tissue damage and degeneration that often require treatments to speed up the regeneration, repair, and/or replacement of the damaged tissue [1,2]. Patient-to-patient rejection, cross-infection risk, and limited donor availability were the major drawbacks facing the previously established methods (autograft, allograft, and xenograft), hence the need for alternative treatment options [3–5].

Tissue engineering (TE), an interdisciplinary field of study harnessing the knowledge of biology, biochemistry, clinical medicine, material and pharmaceutical sciences, and engineering, is one of the alternative options that involves the use of cells, mostly seeded on a three-dimensional (3D) carrier material (the scaffold) with appropriate growth factors to mimic the extracellular matrix (ECM) of the native tissue [3,6]. The success of TE is tightly connected to

**Data Availability Statement:** All relevant data are within the manuscript and its Supporting Information files.

**Funding:** Dr. Saiful Izwan Abd Razak received a research grant number 01K55 from Universiti Teknologi Malaysia for characterization samples.

**Competing interests:** he authors have declared that no competing interests exist.

an appropriate scaffold that enables easy cell attachment and adequate energy transfer for the cells to proliferate and differentiate [6–8]. Bone tissue engineering (BTE) is an essential aspect of TE and a promising alternative to the traditional treatment methods for critical bone defects due to trauma, infection, and tumor resection. It relies mainly on a bioactive scaffold with sufficient mechanical integrity to tolerate the bone remodeling process [6,9].

Advancement in material science and engineering has led to unveiling the potential application of polymeric biomaterials as scaffolds for TE. This is due to the tunability of their properties to resembling the ECM of a native tissue [10], as well as biodegradability and biocompatibility [11]. Bacterial cellulose (BC) is one of the explored polymeric biomaterials in biomedicine [12,13] due to its fascinating properties such as excellent tensile strength, high purity, degree of polymerization, and crystallinity index [14–16]. While a native BC lacks sufficient bioactivity, and osteoconductivity as a BTE scaffold, its hydroxyapatite (HA) composite was found to support *in vitro* osteoblast cell attachment, proliferation, and alkaline phosphatase (ALP) expression [17–19]. Composite scaffolds of HA with other polymeric biomaterials have also been reported to support cell attachment, proliferation, and differentiation [20–23].

Hydroxyapatite (HA) is an inorganic calcium phosphate mineral ($Ca_{10}(PO_4)_6(OH)_2$ found to constitute almost 50% (by weight) of a bone [24,25]. It is a well-known mineral for developing bioactive scaffolds for BTE due to its outstanding osteoinductive, osteoconductive, and cell adhesive potentials [26–28]. However, the nonuniform dispersibility and low nucleation of HA on the BC's surface due to insufficient functionality are still challenges [29].

While attempts have been made to improve the bioactivity and poor cell attachment associated with cellulose scaffolds through the incorporation of HA [4,6,18,30–33], the nucleation of HA is said to be dependent on the material's surface chemistry. It is established that the hydroxyl (–OH) groups of cellulose have a very poor HA induction compared to carboxyl (–COOH) groups [34,35], which could be the basis for the low HA nucleation leading to poor cell attachment on the BC's surface. We therefore hypothesized that the CA-modification could introduce more carboxyl groups, which in turn enhances the HA nucleation ability and improves the modified BC's (MBC) bioactivity sufficient to support osteoblast cell growth and proliferation.

Benefits of the biomimetic synthesis of HA in simulated body fluid (SBF) include cost-effectiveness and eco-friendliness. Moreover, HA can be uniformly deposited on a material's surface without heat treatment, particularly if the material has a conducive surface chemistry [36]. To our knowledge, surface modification with CA and hydroxyapatite biomineralization of the CA-modified BC in simulated body fluid (SBF) is less reported. Here, we synthesize the HA crystals on the modified BC (MBC) samples through the SBF immersion method. The biomineralized MBC (BMBC) samples were characterized by attenuated total reflectance Fourier transform infrared (ATR-FTIR) spectroscopy, X-ray diffraction (XRD), field emission scanning electron microscopy (FE-SEM), and thermal gravimetric analysis (TGA). The BMBC samples have shown enhanced HA nucleation ability and improved physicochemical and mechanical properties compared to the unmodified samples. Moreover, improved biocompatibility and bioactivity were observed for the BMBC to human osteoblast cell lines based on the MTS, Trypan blue dye exclusion (TBDE), and attachment assays.

## Materials and methods

Bacterial cellulose (BC) sheets were purchased from a local *Nata de Coco*-producing company (Happy Alliance) in Malaysia. Citric acid monohydrate powder ($C_6H_8O_7.H_2O$), sodium hydroxide (NaOH) pellets, MTS (3-(4,5 dimethylthiazol-2-yl)-5-(3-carboxymethoxyphenyl)-2-(4-sulfophenyl)-2H-tetrazolium) reagent, glutaraldehyde (25%), and ethanol (99.5%) were

purchased from Sigma Aldrich. Sodium chloride (NaCl), sodium hydrogen carbonate (NaHCO$_3$), potassium chloride (KCl), di-potassium hydrogen phosphate trihydrate (K$_2$HPO$_4$·3H$_2$O), magnesium chloride hexahydrate (MgCl$_2$·6H$_2$O), hydrochloric acid (HCl), calcium chloride (CaCl$_2$), and sodium sulphate (Na$_2$SO$_4$), and tris(hydroxymethyl)amino-methane (CH$_2$OH)$_3$CNH$_2$) were all purchased from Merck. Dulbecco's Modified Eagle Medium (DMEM), Fetal Bovine Serum (FBS), penicillin-streptomycin, Trypsin-EDTA solution (TrypLE™ Express), Trypan blue dye, and phosphate buffered saline (PBS) were purchased from Gibco Life Technologies, USA.

## Bacterial cellulose modification

The citric acid modification and characterization of the BC were as reported in our published work [37]. However, for the fact that the study is targeted towards biomedical application, modified samples showing high water absorption/swelling rate were selected for the biomineralization process. Briefly, purified BC samples were immersed in two different molar concentrations (0.0375 M and 0.075 M) of citric acid (CA) solution in ion-exchanged distilled water (diH$_2$O). Samples were allowed to stand for 24 h at 45°C, then cured at 140°C for 2 h. Another BC sample was treated under the same condition with diH$_2$O only; this served as the control sample. All samples were then removed and rinsed with diH$_2$O until the pH reached 5–6. After this, samples were tagged as BC (pristine), MBC0.03 (0.0375 M), and MBC0.07 (0.075 M) and freeze-dried for characterization [37].

## Synthesis and characterization of HA on BC

For hydroxyapatite (HA) synthesis on both modified and pristine BC, SBF solution was prepared according to Kokubo and Takadama, 2006 [38]. Samples were soaked and incubated in the SBF at 37°C for 1, 7, 14, and 21 days. The SBF solution was changed every 48 h to maintain an optimal ion concentration within the solution. Samples were then removed and rinsed gently with diH$_2$O, then tagged as either BC (pristine) or BMBC (biomineralized BC) based on the soaking time (Table 1). Samples were either freeze-dried or left swollen for further analysis. Freeze-dried samples were subjected to FTIR, XRD, FE-SEM, and TGA, while the swollen samples were subjected to compressive strength testing. Both pure and modified soaked samples were tagged based on their soaking times, as shown in Table 1. Samples showing better HA nucleation based on the FTIR results were subjected to further characterization and were also used for in *vitro* biocompatibility assays.

## Fourier transformed infrared (FTIR) spectroscopy

Using an FTIR spectrophotometer (Model: PerkinElmer-Frontier™, L1280044, Waltham, MA, USA) equipped with an attenuated total reflection system (ATR-FTIR) as in [39], samples were scanned in a wavelength range of 4000 to 650 cm$^{-1}$ and 4 cm$^{-1}$ resolutions. The spectra obtained were plotted as intensities against a wavenumber graph.

Table 1. Samples description based on the simulated body fluid (SBF) soaking period.

| SBF Soaking Period | Sample Description | | |
| --- | --- | --- | --- |
| | **BC** | **MBC0.03** | **MBC0.07** |
| 1—day | BC-S1 | BMBC0.03-S1 | BMBC0.07-S1 |
| 7—days | BC-S2 | BMBC0.03-S2 | BMBC0.07-S2 |
| 14—days | BC-S3 | BMBC0.03-S3 | BMBC0.07-S3 |
| 21—days | BC-S4 | BMBC0.03-S4 | BMBC0.07-S4 |

### X-ray diffraction (XRD)

An X-ray diffractometer (Model: Rigaku SmartLab, USA) with CuKα radiation wavelength ($\lambda$ = 0.154 nm) operated at 40 kV and 30 mA was used for the XRD analysis. Scans were made between angles $2\theta$ of 10˚ to 60˚ at a speed of 3˚/min.

### Field-emission scanning electron microscope (FE-SEM) with energy dispersive x-ray (EDX)

A high-resolution FE-SEM machine (Hitachi SU8020 UHR, Japan) was used for the morphological and microstructural analysis of the extent of HA nucleation on samples after being soaked in SBF for 7 days. Freeze-dried samples were sputter-coated with a thin layer of platinum and scanned at a voltage of 2.0 kV at different magnifications. As for the cross-sectional morphology, samples were broken under a liquid nitrogen to avoid fiber deformations. The elemental quantification data was also obtained from the energy dispersive x-ray (EDX) system.

### Thermal gravimetric analysis (TGA)

A thermal analyzer (Shimadzu DTG-60H, Japan) was used to evaluate the thermal stability of the BMBC. A freeze-dried film weighing 27 mg ± 2 mg from each sample in a platinum pan was heated between 30˚C and 900˚C at a heating rate of 10˚C/min under a nitrogen flow rate of 100 ml/min [40]. Samples were held at 130˚C for 10 minutes to release any absorbed moisture. The weight loss upon heating and the corresponding temperature were obtained from the analyzer, normalized as percentage weight loss (%) and plotted against the corresponding temperature (˚C) [41].

### Compressive strength

A universal testing machine (Instron 8874, Illinois Tool Works Inc., Norwood, MA, USA) was used for the compressive strength testing. Wet and swollen samples (20 x 20 mm) were tested at room temperature with a constant crosshead speed of 1.0 mm/min and 25 N load cell. At least five (5) rectangular samples were tested for each treatment. The compressive modulus was determined from the 0.2% offset at the linear region of the stress-strain curve for each sample [42], and the result was presented as the mean standard deviation.

### *In vitro* biocompatibility test

For a material (natural or synthetic) to be regarded as biocompatible, its response to the biological system must be evaluated. A biocompatible material should have the ability to integrate with living tissues or cells without causing local or systemic adverse effects [43,44]. Our aim here is to evaluate the biocompatibility of the BMBC on the human fetal osteoblast (hFOB 1.19 ATCC® CRL 1137™) cell line. This was performed through the MTS, TBDE, and attachment assays.

**MTS assay.** Wet and swollen biomineralized MBC (BMBC) and the unmodified BC samples (10 x 10 mm) were washed with PBS and UV sterilized in the BSC cabinet for 30 minutes on each side. Samples were then immersed in 1 mL of CDMEM in a 24-well plate at 37˚C and 5% $CO_2$ for 24 h, after which the medium was aspirated and seeded with 1 mL of 5 x $10^5$/mL of human fetal osteoblast (hFOB) 1.19 (ATCC® CRL 11372™) cell suspension in a fresh CDMEM and incubated at the same condition for 3, 5, and 7 days. Wells with cells seeded in CDMEM without the test sample were considered the controls. MTS reagent (100 μl) was then added to all wells except the blank (media only), wrapped in aluminum foil, and incubated at

37°C with 5% $CO_2$ for 3–4 h. 200 µl from each assay well was aspirated and transferred into a 96-well plate, where the absorbance was measured using a microplate reader (ELISA Microplate Reader, Epoch, Biotech) at 490 nm [45]. The percentage (%) cell viability was obtained from the average optical density readings [31] and compared with the control.

**Trypan blue dye exclusion (TBDE) assay.**   Biomineralized BC (BMBC) and unmodified BC samples (10 x 10 mm) were rinsed in 1 x PBS solution for 15 minutes and UV sterilized for 30 mins on each side prior to the experiment. Samples were then immersed in CDMEM for 24 h, after which the old medium was aspirated and replaced with 1 mL of cell suspension in CDMEM. Wells with cells seeded in CDMEM without the test sample were considered the controls. The culture was incubated at 37°C and 5% $CO_2$ for 3, 5, and 7 days, then detached and suspended in CDMEM before centrifugation. The assay was performed according to [46], where an aliquot of the cell suspension was centrifuged at 125 g for 5 minutes, and the supernatant was discarded and then resuspended in 1x PBS. Cells in PBS were mixed with the same volume of 5% trypan blue dye, loaded on the Neubauer chamber, observed using an inverted optical microscope, and counted. The percentage cell viability was calculated using Eq 1 below.

$$Cell\ viability\ (\%) = \frac{total\ no.of\ viable\ cells\ per\ ml\ of\ aliquot}{total\ no.of\ cells\ per\ ml\ of\ aliquot} \times 100 \qquad (1)$$

**Cell adhesion assay.**   Wet BMBC samples measuring 30 x 30 mm were cut in triplicate, washed in PBS for 10 min, and UV sterilized for 30 min on each side. Samples were immersed in CDMEM on a 6-well culture plate and incubated at 37°C with 5% $CO_2$ for 24 h. The old medium was then aspirated and replaced with a 3 mL cell suspension of 1 x $10^5$/mL, incubated at the same condition for 3 days. Samples were then rinsed in PBS, fixed with 2.5% glyceralde-hyde at room temperature [47], and dehydrated using 10%, 20%, 40%, and 60% graded alcohol for 15 mins each, followed by 80%, 90%, and 100% for 30 min each [48]. Finally, the samples were air-dried, placed on studs, and sputter coated with platinum, then viewed under the FE-SEM system [31].

## Results and discussion

Integration of BTE scaffolding materials with the bone is strongly associated with the hydroxy-apatite layer, which is believed to enhance material adhesion to the bone [49]. Using the biomimetic HA synthesis in SBF, which takes advantage of the chemical interaction between $Ca^{2+}$ and $P^{5+}$ ions under controlled pH and temperature [50], the work aims to improve the MBC's bioactivity. Additionally, the physicochemical characteristics and osteoblast cell attachment and proliferation potential of the biomineralized MBC (BMBC) were assessed, with the findings reported below. The author's future research proposal is to further demonstrate the bone regeneration potential of the BMBC by evaluating biological markers of bone formation and regeneration, such as proteins or RNA.

### Fourier transformed infrared (FTIR) spectroscopy

The FTIR spectra presented in Fig 1 depict a comparison between the unsoaked and soaked pure samples and the soaked modified samples at different soaking times. Plate (a) compares the unsoaked and soaked pure BC samples. The slight differences observed in the peaks at 3680–2660 $cm^{-1}$, which are mostly due to OH-stretching vibrations of water molecules [51], could be as a result of differences in moisture content. Peaks at 1630 $cm^{-1}$ due to CH-stretching vibrations tend to be more intense on the soaked than on the unsoaked samples, which

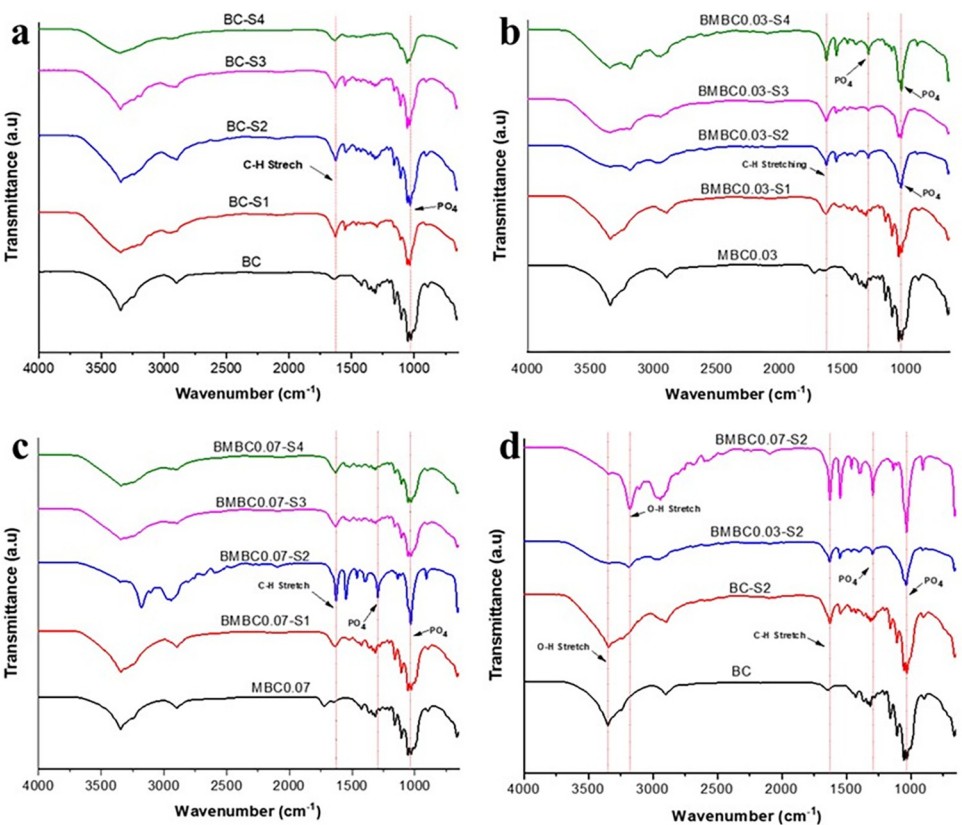

**Fig 1.** Comparative FTIR spectra for (a) pure soaked (BC-S1 to BC-S4) with pure unsoaked (BC), (b) modified soaked samples at different soaking times (BMBC0.03-S1 to BMBC0.03-S4) with modified unsoaked sample (MBC0.03), (c) modified soaked at different soaking times (BMBC0.07-S1 to BMBC0.07-S4) with modified unsoaked samples (MBC0.07), and (d) unmodified and modified 7-day soaked samples (BC-S2, BMBC0.03, and BMBC0.07-S2) with unmodified unsoaked sample (BC).

could be the influence of hydrogen atoms on HA bonding to the carbon atoms on BC. Peaks ascribed to phosphate ($PO_4$) groups of the HA at 1290–034 $cm^{-1}$ [30,52] seem to emerge only on the BC-S4 sample (soaked for 21 days), which indicates that HA nucleation was very low on pure BC samples owing to the lesser carboxylic groups (COO-) on the surface [35].

Plate (b) compares the modified BC samples (MBC0.03) at different soaking times with the unsoaked. A similar scenario to plate (a) can be observed here with respect to the OH and CH-stretching vibrations. All soaked samples have shown the characteristic peaks ascribed to the $PO_4$ groups at 1290–1034 $cm^{-1}$ as opposed to the unsoaked sample [53].

Plate (c) shows a comparison between MBC0.07 samples at different soaking times versus the unsoaked sample. Peaks ascribed to the CH-stretching and $PO_4$ groups comparable to those in plate (b) were also present here. A slightly sharp peak at 3100–3600 $cm^{-1}$ that appears on the BMBC0.07-S2 only is attributed to hydrogen-bonded OH-stretching vibration. The signal is rather broad, possibly due to the restriction of hydrogen bonding from a lack of molecular contact [54].

In all the groups, samples soaked for 7 days seem to have $PO_4$ peaks comparable to those soaked for 21 days, possibly because they can attain a maximum HA nucleation in SBF even at 7 days and were selected as the best samples. Plate (d) is therefore a comparison between the selected samples (7 days soaking) from all groups versus the pure BC. The soaked pure BC also displays the CH-associated peak like that of the CA-treated samples but a low intense $PO_4$

peak like the pure unsoaked sample. It is essentially evident here that the CA modification has influenced BC's fiber bioactivity to reach the maximum HA nucleation at 7 days.

## X-ray diffraction (XRD)

The XRD patterns of the soaked modified samples, unsoaked pure BC, soaked pure BC, and hydroxyapatite (RRUFF ID: R060180) were presented in Fig 2. It can be observed from the XRD patterns that the amorphous peaks associated with BC 2θ = 14.4˚ tend to disappear while the crystalline peaks tend to emerge as the HA crystals increase. Furthermore, the crystalline peaks associated with BC at 2θ = 22.6˚ appeared in almost all the samples, but with reduced intensity as the CA concentration increased. This could be due to the increased amount of HA crystals with increasing CA concentration covering the BC fiber since the soaked, unmodified BC sample still displayed the typical peaks of BC even though it also has some HA-associated peaks [53].

In contrast, the soaked modified samples appear to have peaks that match the HA peaks from the RRUFF database (http://rruff.info) (ID: R060180). The appearance and intensification of HA associated peaks (10.8˚, 25.9˚, 31.7˚, 40.6˚, 45.4˚, and 56.4˚) [55,56], as well as the lowering of BC associated peaks [30], demonstrate the presence of HA within BC fiber

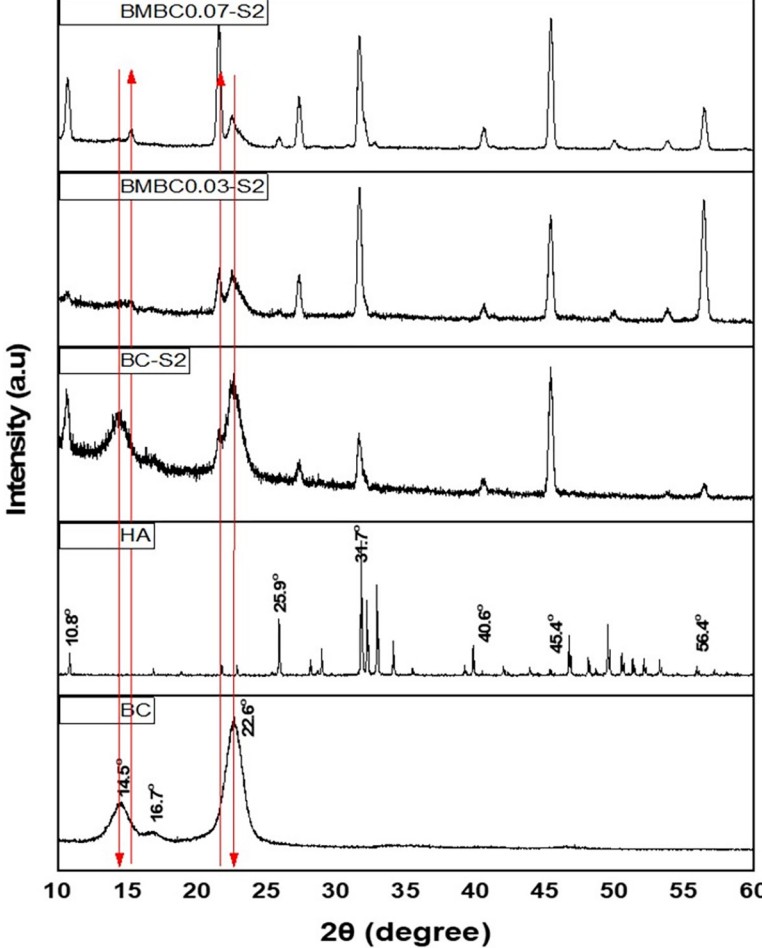

**Fig 2. XRD diffraction patterns of BC, HA, BC-S2, BMBC0.03-S2, and BMBC0.07-S2.**

networks. According to the findings, the synthesized HA is crystalline and expected to be highly resorbable, making it suitable for biomedical applications [57].

## Field-emission scanning electron microscope (FE-SEM) and energy dispersive x-ray (EDX)

The FE-SEM images in Fig 3 displayed the extent of HA nucleation on the soaked modified samples of the same soaking time in comparison to the soaked and unsoaked pure BC samples. A zoom (60.0k) into the surface images revealed an uncovered fiber network typical of BC in the unsoaked and soaked pure samples (BC and BC-S2), respectively, in contrast to the HA crystal-covered fibers in the soaked modified samples (BMBC0.03-S2 and BMBC0.07-S2). It is also noteworthy that the BC fiber still retains its interconnected porous network, especially for the CA-treated samples having the highest HA nucleation. A similarity can also be observed on the cross-sectional images. The unsoaked pure sample (BC) seems to have a clean fiber geometry with interconnected pores, while the soaked pure sample has some traces of HA within its fiber (closer to the periphery of the film), making it a bit compact.

On the other hand, the soaked modified samples (BMBC0.03-S2 and BMBC0.07-S2) showed much denser HA crystals penetrating deep into the fiber. Since the HA nucleation on the modified samples seemed to increase with the increase of CA concentration, it can be presumed that the CA treatment has improved the rate of HA nucleation on the BC, which is much more likely due to the additional carboxyl (COO-) groups on the fiber surface [34,35,58]. The microstructural characteristics shown here correspond with FTIR data and are therefore expected to yield a conducive surface for osteoblast attachment and proliferation.

**Energy dispersive x-ray.** It can also be observed from the EDX elemental spectral peaks in Fig 4 that the unsoaked pure sample (BC) does not contain any trace of either calcium (Ca) or phosphorus (P), while the soaked pure (BC-S2) and modified samples (BMBC0.03-S2, BMBC0.07-S2) all have both peaks but with varying intensities. All, including the unsoaked pure sample, have shown sodium (Na) peaks, which could be due to either the sodium

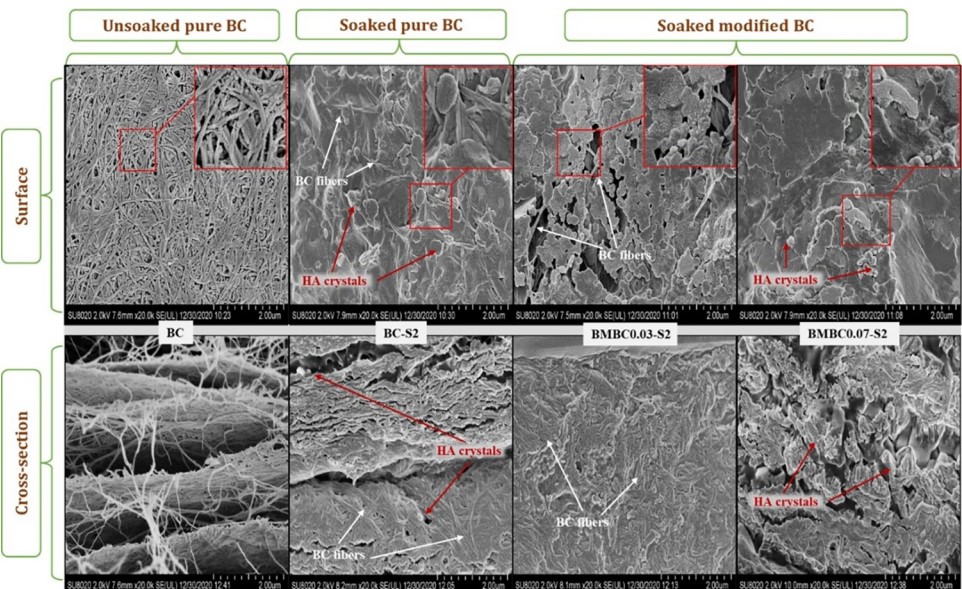

**Fig 3. Surface and cross-sectional FE-SEM images of unsoaked pure (BC), soaked pure (BC-S2), and soaked modified (BMBC0.03-S2 and BMBC0.07-S2) samples.**

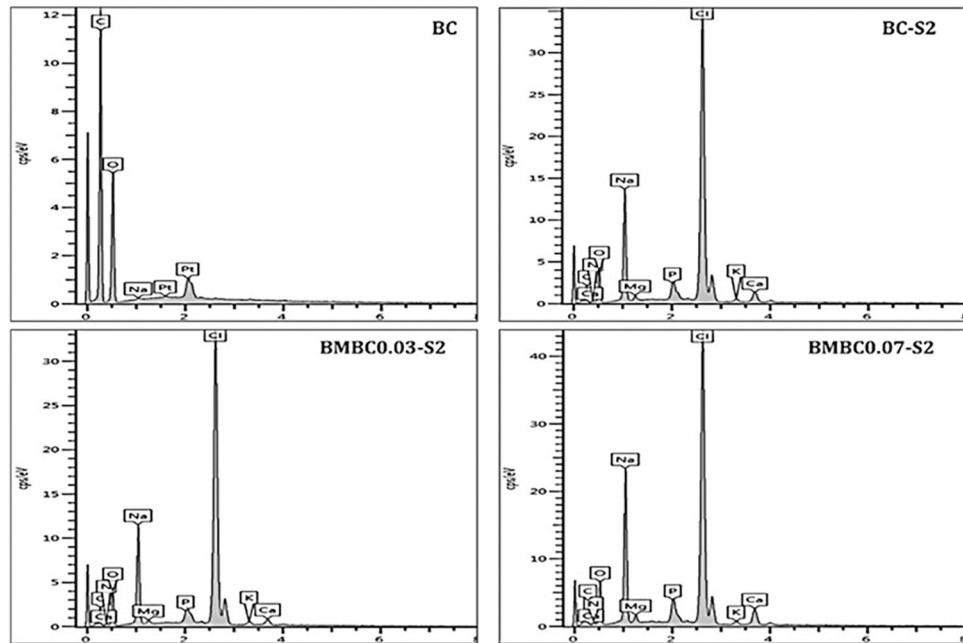

**Fig 4. Elemental maps of the unsoaked pure, soaked pure, and soaked modified samples obtained by EDX analysis.**

hydroxide (NaOH) used for sample purification or the sodium contents of the SBF. All other elements featured in the EDX peaks of the soaked samples, including the pure, could also emerge from the SBF solution [59].

Consequently, for a biocompatible material such as BC, having a long history of being in use for biomedical applications, the enhanced HA nucleation ability will surely improve the bioactivity and widen its applicability in many areas of biomedicine. In BTE specifically, the HA bonding ability of a material is an important prerequisite for it to be used as BTE scaffolding material. According to Kokubo et al. [38], the bone-bonding ability of a material can be predicted by its ability to nucleate HA on its surface in SBF.

### Thermal gravimetric analysis (TGA)

The thermogravimetric curves in Fig 5 were to show the thermal behavior of the unsoaked and soaked modified samples after the HA mineralization in comparison to the pure BC. Plates (a, b, and c) compare the samples soaked for 7 and 21 days (S2 and S4), respectively, for the pure and modified samples with their corresponding unsoaked samples, while plate (d) is between the soaked pure and soaked modified samples.

Partial decomposition of only about 6% of the weight can be seen between the temperatures of 120 and 300°C for the unsoaked samples, while for the soaked samples, the weight loss is about 18% between 120 and 250°C and could be due to the release of moisture content since HA do not dehydrate completely before decomposition [60]. The substantial weight loss of the soaked samples occurred between 250 and 300°C, while the unsoaked pure sample showed the maximum at temperature between 300 and 392°C. Despite the fact that the soaked samples start to decompose at a lower temperature compared to the unsoaked, the soaked samples still retained their weight (46.87–56.48%) up to a temperature of about 500°C, most likely due to the presence of HA that decomposes at higher temperatures than the BC [61]. It is noteworthy that in all cases, samples soaked for 7 days have shown much similar thermal behavior with

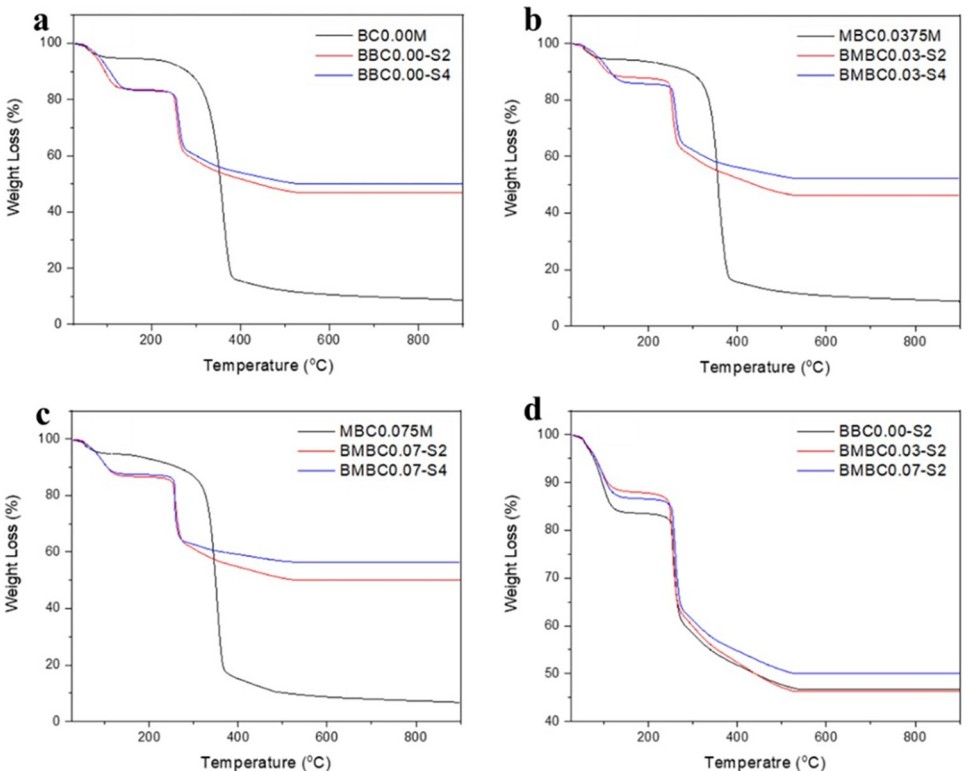

**Fig 5. Comparative TGA curves to show the thermal behavior of the SBF-soaked samples compared to the unsoaked.**

samples soaked for 21 days, indicating that the longer soaking time has less effect on the thermal properties of the samples. This also corresponds with the FTIR results that the MBC can attain its maximal HA nucleation in 7 days.

## Compressive strength

The stress-strain curve for the unsoaked and soaked pure and soaked modified BC samples is shown in Fig 6. All the soaked samples have shown some improved mechanical properties in contrast to the unsoaked pure BC sample. Importantly, the better compressive modulus displayed by the soaked is as expected, taking into account the presence of the apatite crystals filling the void spaces within the BC fiber network [30]. The low modulus seen for BMBC0.03-S2 (although within the accepted limit) [62] could be due to the spongy nature of the samples and low apatite crystals compared to BMBC0.07-S2. The yield strength of all the samples also followed a similar pattern with the compressive modulus, while the fracture point showed not much difference among all the samples. The improved compressive modulus of BMBC reported can be considered an added advantage for its application in BTE, where a minimum compressive modulus of between 2 MPa and 50 MPa is required [62].

## *In-vitro* biocompatibility test

Cell culture and maintenance were performed based on the standard procedure under sterile working and incubation conditions. A class II biosafety cabinet (Thermofisher 1300 series A2) and a $CO_2$ incubator (Binder CB 260) were used all through the preparation and incubation of the cultures. All consumables were of cell culture grade and were used without being

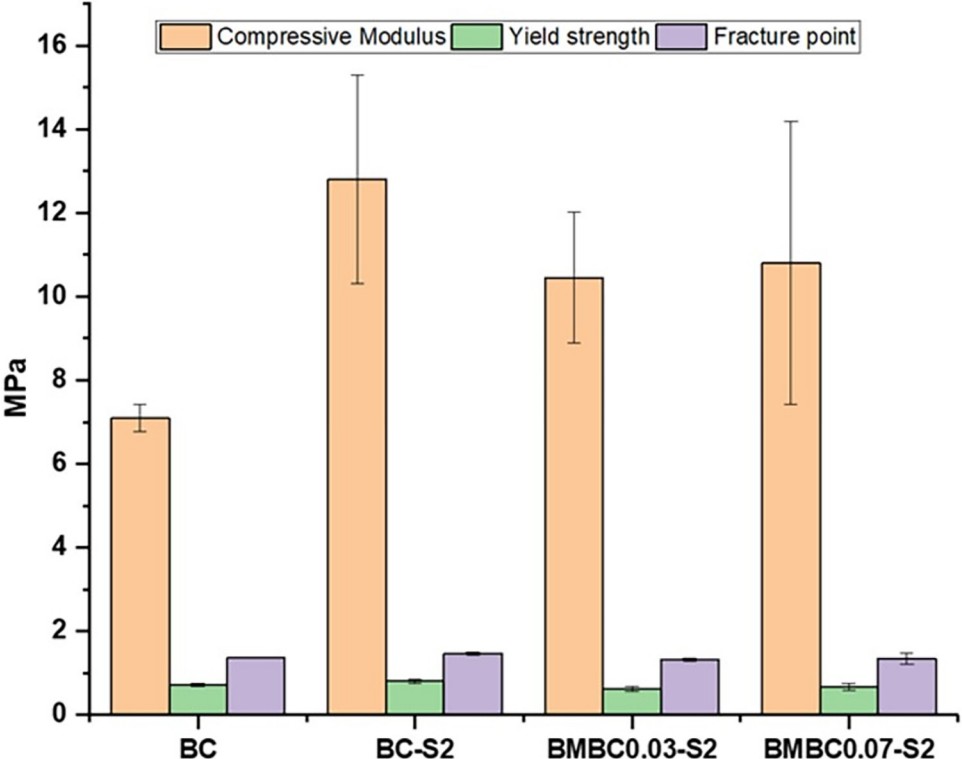

**Fig 6. Compressive mechanical properties for unsoaked pure (BC), soaked pure (BC-S2), and soaked modified samples (BMBC0.03-S2 and BMBC0.07-S2).**

autoclaved. Non-consumables like Schott bottles, scissors, and waste beakers were autoclaved after finishing and before each culture cycle. Surface sterilization was maintained throughout all the experiments using 70% ethanol to prevent cross-contamination. All cell lines used in the research were between passages four (4) and seven (7) and were trypsinized after reaching 70–90% confluency. The images of the hFOB 1.19 cell lines cultured on CDMEM after 1, 2, and 3 days incubation at 37 and 5% $CO_2$ were presented in Fig 7.

**MTS assay.** Cells were cultured and assayed during the proliferative period and after optimizing the seeding density (supplementary data). The results of the assay were statistically analyzed using a one-way ANOVA and the Turkey HSD post-hoc test. The bar charts depicted in Fig 8 represent the viability and proliferation of the hFOB cell lines on the BMBC samples in comparison to the control (cells + media only) and the BC. The significant difference within the same-day culture was found only on the 3-day culture (between the control and BMBC0.03-S2), and there was no significant difference among other samples in the same group. No significant difference was detected within other samples in other culture days (5 and 7 days). Furthermore, there is a significant difference ($P<0.01$) between samples of different culture days (3 and 5 days) and a highly significant difference ($P<0.001$) between the culture at 3 and 7 days (supplementary data), which could be as a result of disturbance at the initial stage of the culture due to extracellular membrane adhesion protein synthesis [29].

It can also be observed that samples did not show any significant difference in cell viability within almost all the groups with the control, except at 3 days. It is also noteworthy that the untreated BC samples did not show a significant difference from the control, and this could be due to the non-toxic nature of the BC [63,64]. The cells could still be viable through the few

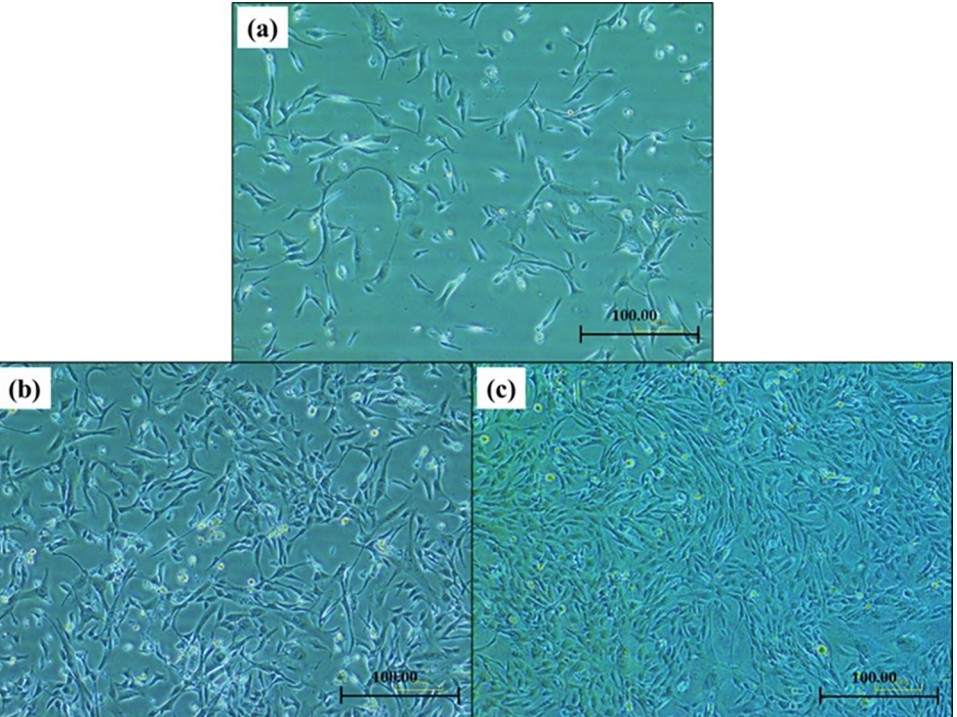

**Fig 7.** Images of hFOB 1.19 cells cultured on CDMEM after (a) one day, (b) two days, and (c) three days. X100.

culture days even though they are poorly attached to the BC, as evidently seen in Fig 10. Furthermore, the CA modification and HA mineralization did not significantly affect the BC's biocompatibility [29] and interestingly enhanced its attachability to hFOB cells, thus could be a good BTE scaffold [65]. The biocompatibility of BC/HA composite hydrogel to fibroblast cell lines has also been suggested in a recent report by Athukorala et al. [66].

**Trypan blue dye exclusion assay (TBDE).**   The results obtained for the microscopic cell viability assay through TBDE are presented in Fig 9. It can be observed from the results that there has not been much cell death on all the tested samples as compared to the control (100%) after the 3, 5, and 7 days of the culture. All the treated samples have shown good cell viability, ranging from 90 to 95% after 3 days, 93 to 97% after 5 days, and 95 to 98% after 7 days of the culture.

This result agrees with some previous reports [18,29,30] and is in support of the non-toxicity of the bacterial cellulose highlighted in Section 1. According to the ISO document (ISO 10993–5), an in vitro cell viability result $\geq$ 80% for a material is proof of its non-toxicity. Therefore, the 90–98% reported here implies that BMBC is non-toxic to the hFOB cell lines and thus can be used for BTE scaffolding.

**Cell adhesion assay.**   Cell adhesion and proliferation are both influenced by the physico-chemical properties as well as the surface functional groups of a material. Osteoblast cells, being adherent, principally rely on cell-surface attachment to survive and proliferate. HA-coated materials are well known as conducive surfaces for the attachment and proliferation of bone cells [52]. The FE-SEM images in Fig 10 were to depict the hFOB attachment and proliferation on the surface of the tested samples. Looking at images of the unsoaked and soaked pure BC samples (BC and BC-S2), respectively, it can be observed that the cells have poorly attached to the surface; as such, there has not been any noticeable proliferation. Cells appeared to be rounded with an impaired membrane and no evidence of cell division.

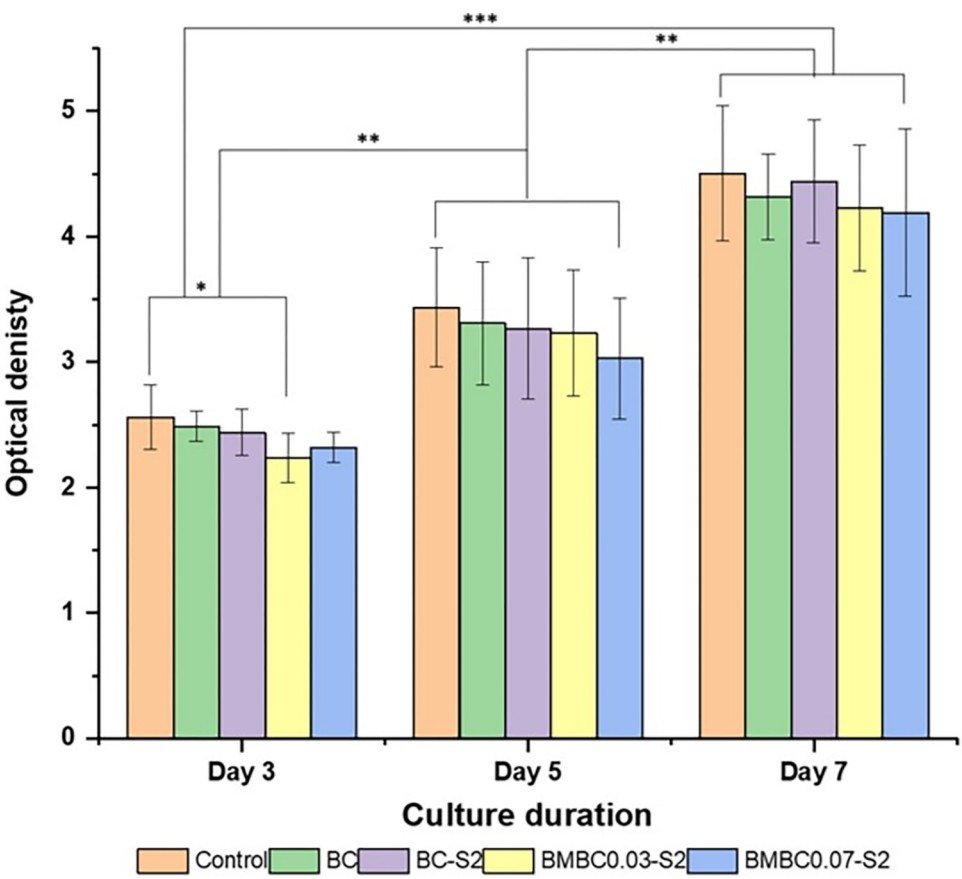

**Fig 8. A bar chart of MTS assay results comparing the hFOB cell proliferation on BC, BC-S2, BMBC0.03-S2, and BMBC0.07-S2 with the control after 3, 5, and 7 days of culture.** *P < 0.05, **P< 0.01, ***P< 0.001, and degree of freedom = 4 (obtained by one-way ANOVA, Tukey HSD test) among all groups.

On the other hand, cells appeared well attached, flattened, and elongated with an intact membrane and filopodia on the surface of the soaked modified samples (BMBC0.03-S2 and BMBC0.07-S2), similar to what was reported in [48]. Both modified sample surfaces displayed densely populated cells with clear evidence of cell division and proliferation. The surface of (BMBC0.07-S2) and, to a lesser degree, (BMBC0.03-S2) is almost covered with viable cells at their proliferative stage with an active filopodia. The enhanced cell attachment and proliferation are probably influenced by the HA deposits, as they tend to be higher on the mineralized samples than the control [18]. This is attributed to the hydroxyl groups, which play a vital role in HA's inherent bioactivity and cellular interaction properties. Composites of hydrogels and HA are advantageous for the synergistic combination of their unique properties (hydrated biocompatible environment and bioactivity, respectively) [67]. A similar result was recently reported for cell attachment on BC and BC/HA [68]. Our result is also consistent with the notion that a pure BC is inherently non-toxic [69]. As such, the cells can proliferate but are poorly attached due to the absence of HA deposits on their surface [68,70]. Despite the small pore diameter associated with native BC, several pore sizes have been achieved through crosslinking/incorporation of porogens within the matrices and are also reported to support osteoblast survival and proliferation [6,71].

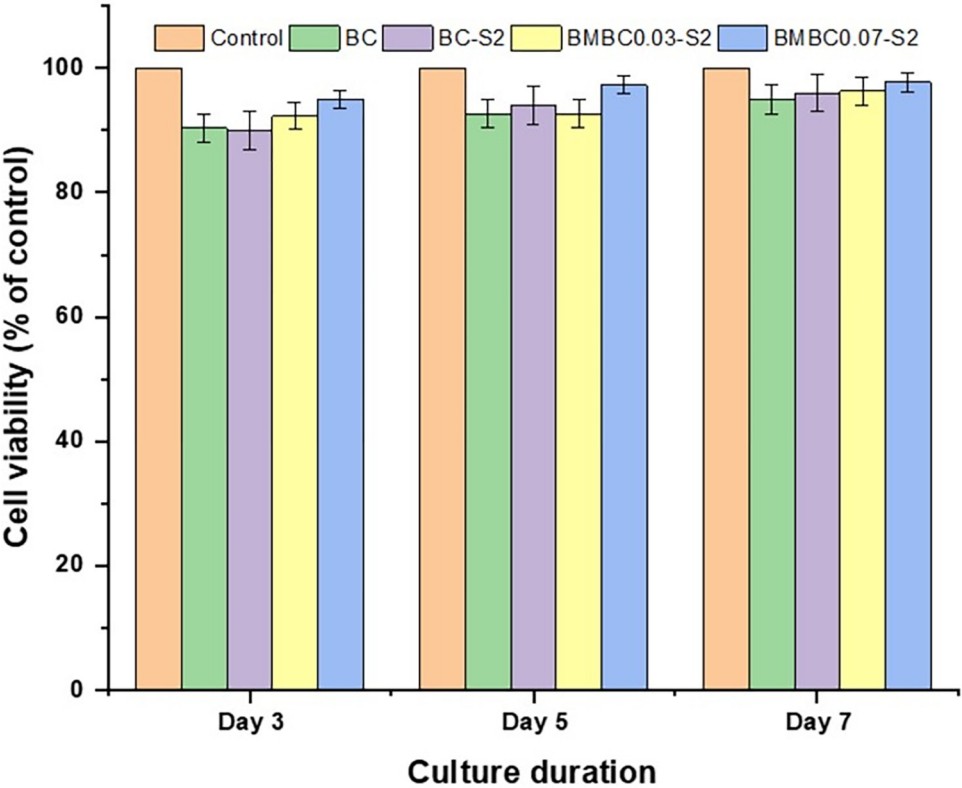

**Fig 9. Trypan blue dye exclusion results presented as percentage cell viability based on the control sample.**

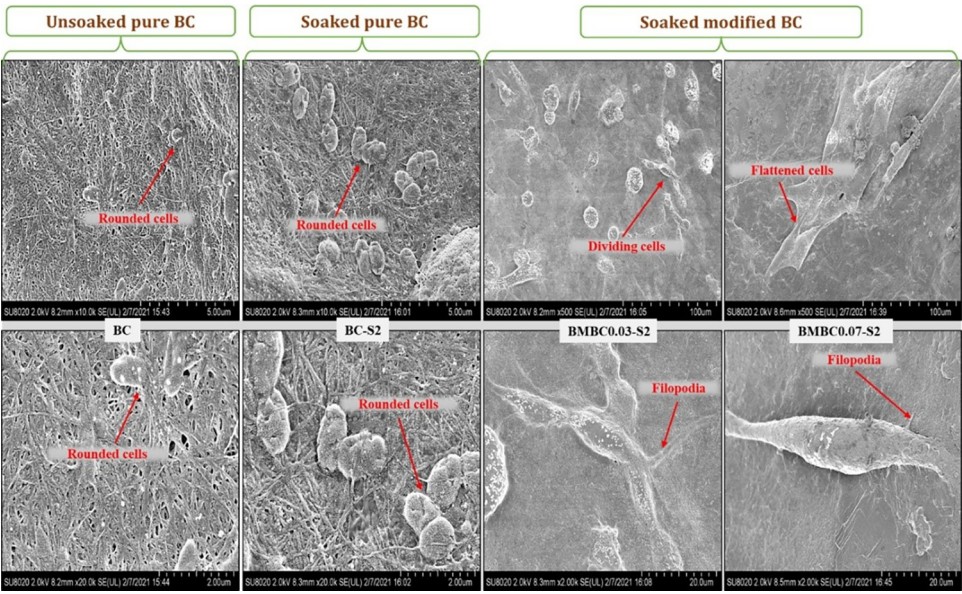

**Fig 10. Surface FE-SEM images of hFOB attachment after 3 days on the unsoaked pure (BC), soaked pure (BC-S2), and soaked modified (BMBC0.03-S2 and BMBC0.07-S2) samples.**

## Conclusion

In summary, a novel biocompatible citrate-modified bacterial cellulose/hydroxyapatite scaffold has successfully been developed. Citric acid crosslinking modification has enhanced the hydroxyapatite biomineralization of the developed scaffold in SBF, which in turn improved the bioactivity and biocompatibility of the BC sufficient to support the attachment and proliferation of osteoblast cells on its surface. The modified BC was found to attain its maximum HA nucleation after 7 days soaking in SBF solution. The apatite formation has also improved the mechanical and physicochemical properties of the BMBC as per the characterizations. Also, on account of the in vitro biocompatibility assessment, the modified and biomineralized samples have demonstrated better cell proliferation and attachment compared to the control (unmodified samples), which is attributive to the intrinsic bioactivity of the HA. The osteoblast cell lines have attached well with filopodia and evidence of cell division after the 3 days of culture. The BMBC reported here could be exploited in biomedicine as potential bone tissue regeneration scaffolding material.

## Supporting information

**S1 Fig. Average growth curve of hFOB cell lines cultured on CDMEM.**
(DOCX)

**S1 Table. Cell initial seeding density and corresponding % confluency/day.**
(DOCX)

**S2 Table. One-way ANOVA and Turkey Post Hoc Multiple comparisons test for 3 days MTS assay.**
(DOCX)

**S3 Table. One-way ANOVA and Turkey Post Hoc Multiple comparisons test for 5 days MTS assay.**
(DOCX)

**S4 Table. One-way ANOVA and Turkey Post Hoc Multiple comparisons test for 7 days MTS assay.**
(DOCX)

**S5 Table. One-way ANOVA and Turkey Post Hoc Multiple comparisons test between groups (3, 5, and 7 days) MTS assay.**
(DOCX)

## Acknowledgments

The authors wish to thank the Faculty of Science, Universiti Teknologi Malaysia, for providing the conducive laboratory environment to support this work.

## Author Contributions

**Conceptualization:** Rabiu Salihu, Saiful Izwan Abd Razak.

**Data curation:** Rabiu Salihu.

**Formal analysis:** Rabiu Salihu.

**Funding acquisition:** Saiful Izwan Abd Razak.

**Investigation:** Rabiu Salihu.

**Methodology:** Rabiu Salihu.

**Project administration:** Shafinaz Shahir.

**Resources:** Mohd Helmi Sani, Nurliyana Ahmad Zawawi.

**Software:** Rabiu Salihu.

**Supervision:** Saiful Izwan Abd Razak, Mohd Helmi Sani, Nurliyana Ahmad Zawawi, Shafinaz Shahir.

**Validation:** Nurliyana Ahmad Zawawi.

**Visualization:** Mohammed Ahmad Wsoo.

**Writing – original draft:** Rabiu Salihu.

**Writing – review & editing:** Mohd Helmi Sani, Mohammed Ahmad Wsoo, Shafinaz Shahir.

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
