## [Decision Letter · Decision Letter 0]

12 Aug 2024

PONE-D-24-23106Citrate-modified Bacterial Cellulose as a Potential Scaffolding Material for Bone Tissue RegenerationPLOS ONE

Dear Dr. Wsoo,

Thank you for submitting your manuscript to PLOS ONE. After careful consideration, we feel that it has merit but does not fully meet PLOS ONE’s publication criteria as it currently stands. Therefore, we invite you to submit a revised version of the manuscript that addresses the points raised during the review process.

We look forward to receiving your revised manuscript.

Kind regards,

Carlos Alberto Antunes Viegas, DVM; MSc; PhD

Academic Editor

PLOS ONE

 [Dr. Saiful Izwan Abd Razak received a research grant number 01K55 from Universiti Teknologi Malaysia for characterization samples.].  

[The authors would like to thank Universiti Teknologi Malaysia, under research grant number 01K55, for providing financial support for characterization samples.]

 [Dr. Saiful Izwan Abd Razak received a research grant number 01K55 from Universiti Teknologi Malaysia for characterization samples.]

5. Please include your tables as part of your main manuscript and remove the individual files. Please note that supplementary tables (should remain/ be uploaded) as separate ""supporting information"" files".

Reviewers' comments:

Reviewer's Responses to Questions

**Comments to the Author**

1. Is the manuscript technically sound, and do the data support the conclusions?

Reviewer #1: Yes

Reviewer #2: No

2. Has the statistical analysis been performed appropriately and rigorously? 

Reviewer #1: Yes

Reviewer #2: N/A

3. Have the authors made all data underlying the findings in their manuscript fully available?

Reviewer #1: Yes

Reviewer #2: Yes

4. Is the manuscript presented in an intelligible fashion and written in standard English?

Reviewer #1: Yes

Reviewer #2: No

5. Review Comments to the Author

Reviewer #1: Review: PONE-D-24-23106

Title: Citrate-modified Bacterial Cellulose as a Potential Scaffolding Material for Bone Tissue Regeneration

The authors present results regarding the synthesis of the HA crystals and the modified BC (MBC) samples through the SBF immersion method. The manuscripts present the characterization of the biomineralized MBC (BMBC) samples by attenuated total reflectance Fourier transform infrared (ATR-FTIR) spectroscopy, X-ray diffraction (XRD), Field emission scanning electron microscopy (FE-SEM), and thermal gravimetric analysis (TGA). Also, the authors provided information on the biocompatibility of the BMBC to human osteoblast cell lines. The biocompatibility of the samples was evaluated through the MTS, Trypan blue dye exclusion (TBDE), and attachment assays.

The overall manuscript is well structured and present useful information about the topic.

Nonetheless, the manuscript needs major improvements before being considered for publication. Please have in mind the following improvements:

1. The abstract should be more concise and state the motivation and the major achievements of the study.

2. The introduction should be improved and the authors should emphasize the novelty of their study compared to existing studies.

3. Also, the author should add the EDX spectra to the manuscript as well as the images of hFOB 1.19 cells cultured on CDMEM which were provided in the supplementary section.

4. The FTIR vibrational bands should be assigned and summarized in a table in the manuscript. Also, the second derivative of the FTIR should be included.

5. The discussions section should be improved. The biological assays should be correlated with the physico-chemical properties of the samples. Also, the authors should provide a possible explanation regarding why according to the in-vitro biocompatibility assessment, the modified and biomineralized samples have demonstrated better cell proliferation and attachment compared to the control.

6. The conclusions section should also be improved.

Reviewer #2: 1- It is necessary to edit the native English language.

2- The image quality should be enhanced.

The manuscript does not mention the innovative aspects of the work.

Please revise the discussion section to include the limitations and suggestions. Additionally, highlight the advantages of your work compared to other materials.

Bacterial cellulose has a significantly small pore diameter. This raises the question of how it can serve as a suitable biomaterial for bone tissue engineering. The challenge lies in the inability to efficiently transfer cells to the central layers. please discussion about this limitation.

6. PLOS authors have the option to publish the peer review history of their article (what does this mean?). If published, this will include your full peer review and any attached files.

Reviewer #1: No

Reviewer #2: No

---

## [Author Response · Author response to Decision Letter 0]

25 Sep 2024

Reviewer 2 

The manuscript has been sent for grammar checking and we believe that the major gramatical errors have been taken care of. Thank you

---

## [Decision Letter · Decision Letter 1]

7 Oct 2024

Citrate-modified bacterial cellulose as a potential scaffolding material for bone tissue regeneration

PONE-D-24-23106R1

Dear Dr. Salihu,

We’re pleased to inform you that your manuscript has been judged scientifically suitable for publication and will be formally accepted for publication once it meets all outstanding technical requirements.

Kind regards,

Carlos Alberto Antunes Viegas, DVM; MSc; PhD

Academic Editor

PLOS ONE

Additional Editor Comments (optional):

Reviewers' comments:

Reviewer's Responses to Questions

**Comments to the Author**

1. If the authors have adequately addressed your comments raised in a previous round of review and you feel that this manuscript is now acceptable for publication, you may indicate that here to bypass the “Comments to the Author” section, enter your conflict of interest statement in the “Confidential to Editor” section, and submit your "Accept" recommendation.

Reviewer #1: All comments have been addressed

Reviewer #2: All comments have been addressed

2. Is the manuscript technically sound, and do the data support the conclusions?

Reviewer #1: Yes

Reviewer #2: Yes

3. Has the statistical analysis been performed appropriately and rigorously? 

Reviewer #1: Yes

Reviewer #2: Yes

4. Have the authors made all data underlying the findings in their manuscript fully available?

Reviewer #1: Yes

Reviewer #2: Yes

5. Is the manuscript presented in an intelligible fashion and written in standard English?

Reviewer #1: Yes

Reviewer #2: Yes

6. Review Comments to the Author

Reviewer #1: The authors have responded as best as they could to my comments. Therefore, I think that the manuscript could be accepted for publication.

Reviewer #2: accept،The authors have addressed all requested items.

7. PLOS authors have the option to publish the peer review history of their article (what does this mean?). If published, this will include your full peer review and any attached files.

Reviewer #1: No

Reviewer #2: **Yes: **yes

---

## [Editor Report · Acceptance letter]

14 Oct 2024

PONE-D-24-23106R1 

PLOS ONE

Dear Dr. Salihu, 

I'm pleased to inform you that your manuscript has been deemed suitable for publication in PLOS ONE. Congratulations! Your manuscript is now being handed over to our production team.

Kind regards, 

on behalf of

Dr. Carlos Alberto Antunes Viegas 

Academic Editor

PLOS ONE